# Design and Test of a Jet Remote Control Spraying Machine for Orchards

**Chi Ma** [1] , **Guanglin Li** [1,*] **and Qiangji Peng** [2]

1   College of Engineering and Technology, Southwest University, Chongqing 400715, China;
    mc2018@swu.edu.cn
2   Academy of Agricultural Machinery Science in Shandong, Jinan 250214, China; pengqiangji@shandong.cn
*   Correspondence: liguanglin@swu.edu.cn; Tel.: +86-023-68251265

**Abstract:** Aimed at issues associated with the poor air supply and poor automatic targeting accuracy of existing orchard sprayers, this paper designs a jet-type orchard remote control sprayer with automatic targeting which is suitable for standardized orchards in hilly and mountainous areas. By optimizing the structure of the diversion box, the uniformity of deposition and penetration ability of the pesticide droplets to the fruit tree canopy are improved, and a uniform wild field distribution is realized simultaneously. An accurate positioning of the fruit tree canopy space orientation is achieved through automatic targeting and azimuthal adjustment systems. When the target is detected, the solenoid valve is controlled to open, and vice versa, and the distance from the nozzle to the fruit tree canopy is adjusted in real time to improve the utilization rate of pesticides. The test results show that the effective range of the jet-type orchard remote control sprayer is no more than 3.5 m, and the maximum flow rate range is 6~6.5 L/min. Within the effective spraying range, the farther the distance is, the higher the automatic targeting accuracy. The pesticide droplets sprayed by the spraying machine have a certain penetration ability, and the uniformity of the droplets is good, which solves solidification problems caused by the penetration of pesticide into the soil. This research provides a reference for jet spraying operation and automatic targeting spraying structure design.

**Keywords:** automatic targeting; hills and mountains; jet-type; sensor; spray

## 1. Introduction

The abundant number of sunshine hours and high air humidity in southern China are especially suitable for the development of the orchard economy. However, due to the restriction of its hilly and mountainous terrain, the quality of automatic spraying of pesticides for fruit trees is not sufficient [1,2], which affects fruit production and quality. At this stage, the mechanized spraying operation has been applied to most of the hilly and mountainous standardized orchards. Compared with the manual spraying operation, its efficiency has been greatly increased, but the spraying quality has not been effectively improved. To solve the abovementioned problems, many scholars have carried out research on the suitable mechanism of hills and mountains. In terms of carrying stability, the orchard self-propelled sprayer, after optimizing the power and chassis [3–5], has strong climbing and obstacle crossing capabilities, which solves the problem of difficulty in entering the garden. Nevertheless, the vehicle bumps caused by the undulation of the road surface in the process of manual driving, can easily make the driver fatigued. The remote control spraying machine [6,7] can effectively reduce the labor intensity of plant protection workers, and the machine has a small size and a long remote control distance. However, due to the lack of target recognition and automatic switching capabilities of the liquid path, the utilization rate of pesticides is low, and the amount of pesticide residues in the soil is large. Other problems still exist, and the quality of spraying operations has not been effectively improved.

At present, there are two main ways to effectively improve the quality of spraying operations. One is to optimize the air supply intensity [8–12], and the other is to improve the accuracy of automatic targeting alignment [13,14]. Among them, the method of optimizing the air supply intensity is relatively simple and usually based on the CFD method to analyze the wind field intensity and flow conditions outside the fan, air duct, and air outlet and change the diversion structure and air outlet mode of the air supply system [15,16]. The half-circle, diffuse type is the most commonly used schematic, but its directional anti-floating effect and auxiliary droplet penetration effect are poor [12]. Optimizing the air duct to achieve jet-type wind can effectively improve the utilization rate of the wind field, but the existing research [17,18] lacks experimental analyses and structural optimization schemes.

In addition to optimizing the air supply device, upgrading hardware, and optimizing algorithm usage, using ultrasonic sensors [19,20], infrared sensors [21], laser scanners [22–24], image processing [25,26], and other sensors to detect the position parameters of fruit trees, the spraying quality of fruit trees can be improved. Although methods such as laser scanners and visual image processing have high target accuracy, the high hardware costs are not conducive to the use and promotion of small- and medium-sized spraying equipment. How to use ultrasonic sensors to accurately target fruit trees while ensuring the quality of spray and increasing the rate of pesticide deposition are key issues that must urgently be resolved in hilly and mountainous plant protection operations.

In summary, designing and optimizing the air supply system and automatic target system are important means to improve the spray quality of the orchard sprayer. Aimed at problems of the poor air supply's effect on the existing spraying equipment in hilly and mountainous orchards and the system's poor automatic targeting accuracy and low pesticide deposition rate, this paper uses the method of analyzing the flow field in the diversion box to guide the design of the air supply system and provides a new technical research idea for the jet orchard sprayer. At the same time, this paper also has developed a jet-type remote control spraying machine suitable for standardized orchards in hilly and mountainous areas. Various indicators reflecting the spraying effect are tested, and the key parameters are optimized in the laboratory in order to achieve high-quality and high-efficiency orchard spraying application demonstration.

## 2. Materials and Methods

According to the previous reports, the fruit trees grown in standardized orchards in hills and mountains are mostly dwarf varieties, and some varieties of fruit trees have a certain degree of canopy closure. Except for the different structure of fruit trees and the different types of pesticides, the spraying methods are basically the same. Therefore, the spraying machine designed in this paper is suitable for any orchard with a planting row spacing of 3~6 m, an unlimited plant spacing, a tree height of 1~3.5 m, and a canopy thickness of 1~2.5 m.

### 2.1. Equipment Structure

The jet-type orchard remote control sprayer mainly consists of a carrying and power unit, a spraying operation unit, an air supply operation unit, and an automatic targeting and position adjustment system; its structure is shown in Figure 1.

The main functions of the above components are as follows:

- The carrying and power unit is mainly responsible for the maintenance of remote control travel and steering.
- The spraying operation department mainly provides power for the atomization and spraying of pesticides.
- The function of the automatic target and position adjustment system is to identify the spatial position of the fruit tree canopy and adjust the atomizing nozzle to the optimal spraying position.

- The air supply operation department mainly provides additional kinetic energy for the sprayed droplets and makes the canopy surface blades swing so that the droplets adhere to the inner chamber.

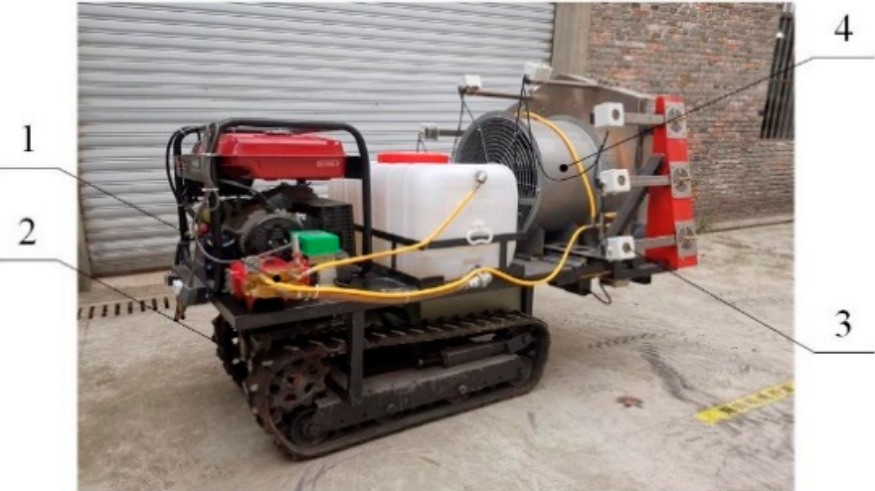

**Figure 1.** The prototype of a jet-type orchard remote control sprayer: (1) spraying operation department; (2) carrying and power unit; (3) automatic targeting and position adjustment system; (4) air supply operation department.

The main technical parameters of the jet orchard remote control sprayer are shown in Table 1.

**Table 1.** Main technical parameters.

| Parameters | Values |
|---|---|
| Boundary dimension (m × m × m) | 1.8 × 1.25 × 1.4 |
| Empty mass (kG) | 850 |
| Equipment power (kW) | 7.25 |
| Travel speed (m/s) | 0~1.5 |
| Maximum drug loading (L) | 300 |
| Lifting range (mm) | 0~300 |
| Horizontal movement range (mm) | 0~300 |
| Fan power (kW) | 0.55 |
| Outlet wind speed range (m/s) | 0~5.8 |
| Spraying power (kW) | 1.5 |
| Effective spraying distance (m) | 3.5 |
| Theoretical spraying volume (L/min) | 6.0~6.5 |

### 2.2. Working Principle

First, the gasoline engine is started, and the orchard remote control sprayer enters a standby state. The operator adjusts the lifting device and the air supply intensity as needed until the nozzles on both sides face the canopy of the fruit tree, and the wind blows the leaves on the surface of the canopy to a swing state. At the same time, the control system begins to measure the speed of the vehicle.

During operation, the ultrasonic sensor sends out a detection signal, which is received by the fruit tree canopy after diffuse reflection. When the sensor detects that the distance of the fruit tree is within the set interval, the controller delays according to the speed of the vehicle until the nozzle is facing the canopy of the fruit tree, the solenoid valve is opened, and the spraying operation is performed. At the same time, the controller sends a signal to adjust the nozzle and the horizontal position of the air outlet. When the sensor detection signal is not within the set interval, the solenoid valve is closed until the sensor detects the

signal within the set interval again, and the system restarts. The working principle of the jet orchard remote control sprayer is shown in Figure 2.

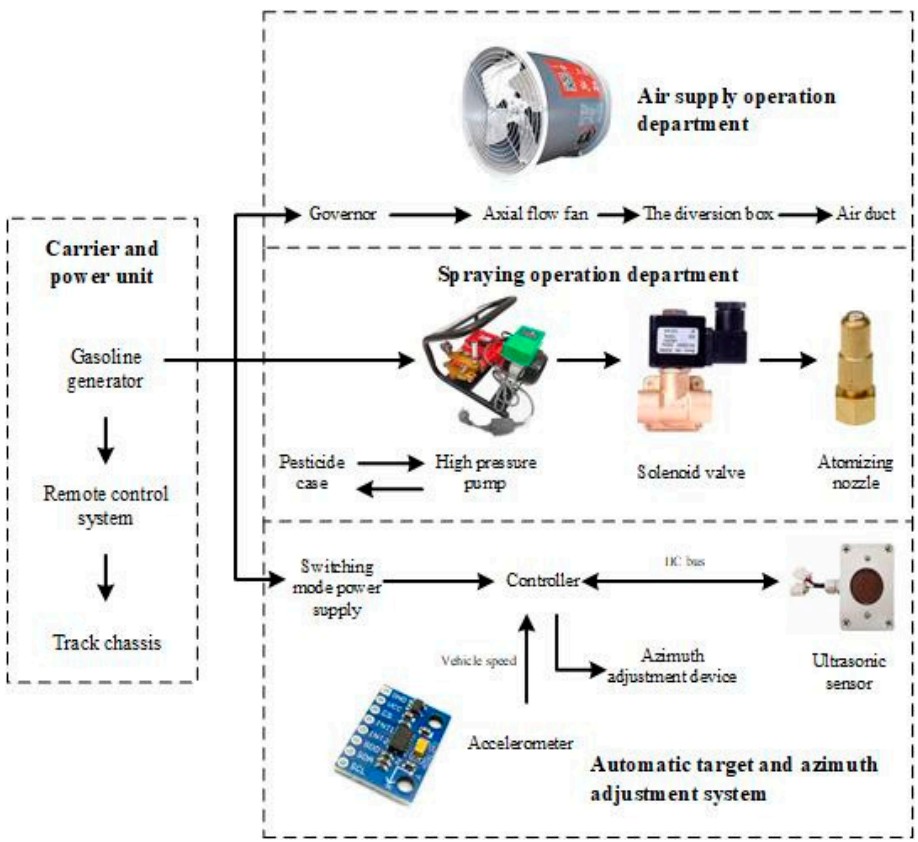

**Figure 2.** Working principle.

*2.3. Design of Carrying and Power Unit*

In China, hilly orchards are generally planted on gentle rises with slopes of 10°~25°, where the road surface is undulating. To improve the climbing ability and carrying stability of the sprayer, the carrying equipment in this study adopts a crawler chassis with independent tracks on both sides equipped with a 2.2 kW AC motor and reducer to realize chassis travel and differential steering functions. A worm gear reducer with a reduction ratio of 1:50 is used here. A 220 V/2.2 kW inverter is used to control the starting current and reduce the starting inertia. In addition, by adjusting the inverter output frequency, the vehicle speed can be changed. The F21-E1B controller is used to realize the remote control of the chassis function. An LX700 linear actuator motor is used to realize the overall lifting of the structure of the sensor and the air supply unit. In terms of power, a Jialing JL10000E gasoline generator is used for the overall power supply, with a power supply voltage of 220 V and a rated power of 8 kW.

In the carrying and power unit, a lifting platform with a size of 800 mm × 1250 mm is designed at its tail. In the initial position, the lifting platform is flush with the fixed platform of the unit. The platform is driven by a linear actuator motor for semi-automatic lifting operations, and the maximum ascent height is 300 mm, which improves the adaptability of the sprayer to fruit trees of different heights. This study established a three-dimensional model to determine the position of the center of mass of the fan under four conditions of no load and full load in the initial state and the highest lift state of the fan and calculated the carrying stability of the jet-type orchard remote control sprayer. The relevant parameters are shown in Table 2.

**Table 2.** Stability calculation parameters.

| Parameters | Initial State | | Maximum Lifting State | |
|---|---|---|---|---|
| | Empty Load | Full Load | Empty Load | Full Load |
| Distance from centroid to rear support point $S_r$ (mm) | 438 | 421 | 438 | 425 |
| Distance from centroid to front support point $S_f$ (mm) | 617 | 633 | 617 | 629 |
| Distance from centroid to ground $S_g$ (mm) | 422 | 514 | 475 | 553 |
| Longitudinal overturning condition $S_r/S_g$ | 1.04 | 0.82 | 0.92 | 0.77 |
| Lateral overturning condition $a/2S_g$ | 0.95 | 0.78 | 0.84 | 0.72 |

$\delta$ is the slip rate, and the value is 0.7. $a$ is the track width of the track chassis, and the value is 800 mm.

According to Table 2, the longitudinal and lateral tilting conditions of the sprayer under the four boundary conditions are both greater than the slip rate, which meets the stability design requirements.

### 2.4. Design of Spraying Operation Department

There are many kinds of orchards in hills and mountains, and the rows and spacing of different varieties of fruit trees are not the same, but that of most fruit trees is between 3~6 m and 1.5~4 m. To meet the spraying needs of various types of orchards, this study connects eight sets of solenoid valves and atomizing nozzle components in parallel to achieve 180° spraying. The spatial layout and liquid path transmission are shown in Figure 3.

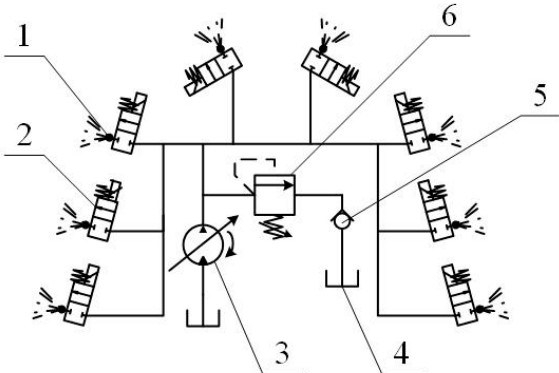

**Figure 3.** Diagram of hydraulic system: (1) atomizing nozzle; (2) solenoid valve; (3) high pressure pump; (4) pesticide box; (5) one-way valve; (6) relief valve.

This part of the power source is still a gasoline generator, which can be converted to 24 V by switching power supply. Table 3 shows that, when the working pressure of the jet orchard sprayer is 1.6 MPa, it can meet the working requirements of the spraying operation department. The kinematic viscosity of water at 25 °C water temperature is $\gamma = 0.897 \times 10^{-6}$ m²·s, the critical Reynolds number Rec = 2300, and the effective spray rate is Qw = 360 L/h under the working pressure of 1.6 MPa. When the nozzle is fully opened, then the high pressure flow velocity vh and Reynolds number Reh in the tube are, respectively, calculated as:

$$v_h = \frac{Q_w}{A_h} = 11.04 \text{ m/s} \tag{1}$$

$$Re_h = \frac{v_h d}{\gamma} = 1.05 \times 10^5 > 2300 = Re_c \tag{2}$$

**Table 3.** Main technical parameters of the spraying operation department.

| Part Name | Type | Technical Index | Values |
|---|---|---|---|
| Three-cylinder plunger pump | SN-26 | Pressure range (MPa)<br>Flow range (L/min) | 0.05~3.5<br>14~22 |
| High pressure pipe | PVC | Inner diameter (mm) | 8.5 |
| Solenoid valve | 0927 | Rated voltage (V)<br>Working pressure (MPa) | 24<br>0.1~1.6 |
| Atomizing nozzle | Ceramics | Nozzle aperture (mm)<br>Maximum flow (L/min)<br>Working pressure (MPa)<br>Sprinkling width (°) | 1.2<br>0.75<br>0.6~3.0<br>80 |

In the formula, $A_h$ is the cross-sectional area of the inner hole of the high-pressure pipe in mm$^2$.

Therefore, the pipe flow in the spraying operation department is turbulent. Since the connection of the high-pressure pipe and the nozzle of the pesticide solution has a sudden shrinking pipe structure, the local head loss coefficient $\zeta$, the local head loss $h_j$, and the local pressure loss $P_j$ are, respectively, calculated as:

$$\zeta = 0.5 \times \left(1 - \frac{A_a}{A_h}\right) = 0.49 \tag{3}$$

$$h_j = \zeta \frac{v_h}{2g} = 0.276 \text{m} \tag{4}$$

$$P_j = \rho g h_j = 2.705 \text{KPa} \tag{5}$$

In the formula, $A_a$ is the cross-sectional area of the nozzle hole in mm$^2$.

In summary, the local pressure loss at the connection between the high-pressure pipe and the nozzle is much smaller than the working pressure, its influence on the spraying effect is negligible, and the spraying operation department can work normally.

*2.5. Design of Air Supply Operation Department*

2.5.1. Fan Selection

The air supply unit is composed of an axial fan, a wind box, an air duct, etc., and its functions are as follows [27]: (1) Generate a wind field and drive the droplets to spray to the fruit tree canopy; (2) promote the swing of fruit tree leaves; and (3) enhance the droplet penetration ability. The wind box of most tower air-blown sprayers is composed of a vertical plate and a guide plate. Because it does not have the ability to adjust the vertical position, the height of the wind box and the cross section of the tuyere are large, and the nozzles are evenly installed at the tuyere to reduce the number of droplets in the environment. The trajectory deviates under the influence of wind, but the uniformity of the air supply is poor, the air volume loss is significant, and the kinetic energy of the droplets is insufficient. To increase the outlet wind speed and ensure the balance of the outlet wind speed, this article designs the air supply operation department.

According to the air volume replacement principle and the wind speed end speed principle [11], the air volume demand of the axial flow fan should meet the following conditions [28]:

$$Q = \frac{(1000 \times V \times L \times H)}{K} \tag{6}$$

where $Q$ is the air volume of the fan, m$^3$/s; $V$ is the travel speed of the sprayer, m/s; $L$ is the effective width of the operation, m; $H$ is the height of the fruit tree, m; and $K$ is the airflow attenuation and loss coefficient along the way, generally taken as K = 3~3.5 (Ding et al., 2013). Combining the construction of standardized orchards in hills and mountains and the

design plan of this study, the values of the abovementioned parameters are $V$ = 1~1.5 m/s, L = 3~5 m, and H = 2~4 m, and the calculated $Q \approx$ 1715–10,000 m$^3$/s. Since the calculated air volume demand range is large, the selected fan is as close as possible to reaching the maximum demand to meet the spraying requirements of fruit trees with higher canopy density and improve the ability of pesticide droplets to penetrate the fruit tree canopy. Therefore, the T35-11-5.6 axial flow fan produced by Shaoxing Shangyu Eslite Fan Co. Ltd. (Shaoxing, China), with a rated speed of 1450 r/min, an air volume of 8667 m$^3$/h, a wind pressure of 172 Pa, and a rated power of 0.55 kW at this speed, was selected.

2.5.2. Design and Simulation of Diversion Box

Due to the different internal structures of the diversion box, the intensity of the airflow is attenuated to varying degrees during the process of converging to ejection, and the change in the outlet wind speed is random, which affects the analysis of the spraying effect of the orchard. Therefore, this research designed a diversion box structure with an inlet diameter of 570 mm and outlet diameter of 100 mm facing the deflector. The intersection of the central axis of the inlet and the deflector is the intersection of the deflector, and the three planes intersect at this point. The angles between the planes are 150°, 160°, 170°, and 180°. The diversion box size is shown in Figure 4.

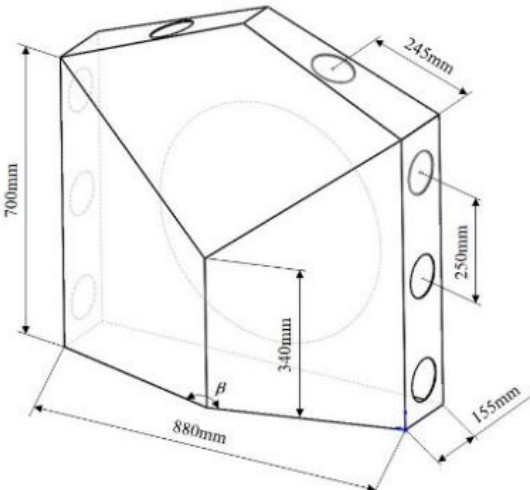

**Figure 4.** Diversion box size chart. The angle $\beta$ is 150°, 160°, 170°, and 180°, respectively.

Under the condition of the inlet wind speed of 10 m/s, the k-ε turbulence model in the Fluent solver is used. A tetrahedral grid is selected for division, and the grid size is 5 mm. The boundary conditions are set to ideal wall surface, the relative atmospheric pressure is 0, the ambient temperature is 300 K, the air condition is the default condition in the fluent solver, and its dynamic viscosity is 1.7894 × 10$^{-5}$ kg·m/s. Under the above configuration, the internal flow field analysis of four kinds of diversion boxes is completed.

As shown in Figures 5–7, as the angle of the baffle increases gradually, the wind force acting on the baffle increases more drastically, and the stability of the baffle structure worsens. The flow field in this model is turbulent, but from the configuration of related parameters and results, it is shown that its Reynolds number is low, which is close to laminar flow. This is the function of the unique convex structure of the diversion box, and its purpose is to effectively use the air volume and distribute it to each air outlet as much as possible, reducing the loss of air volume and wind speed. Table 4 shows the simulation value of the maximum wind speed of each air outlet under the conditions of different angles of the deflector.

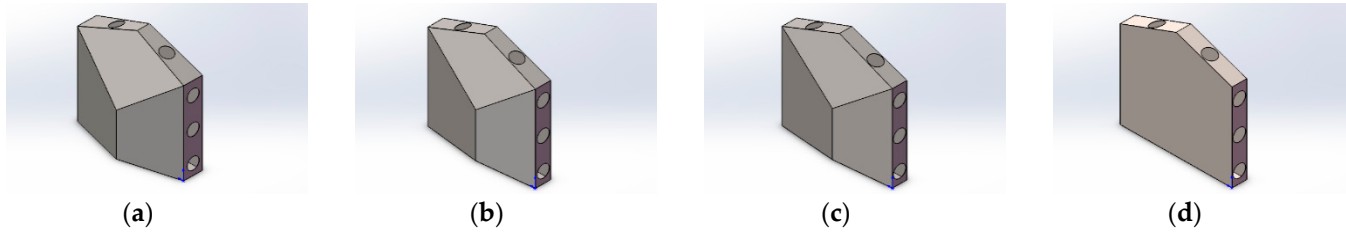

**Figure 5.** Structural model of diversion boxes with different angles of diversion plates. The angles of the deflector in the picture are (**a**) 150°, (**b**) 160°, (**c**) 170°, and (**d**) 180° from left to right.

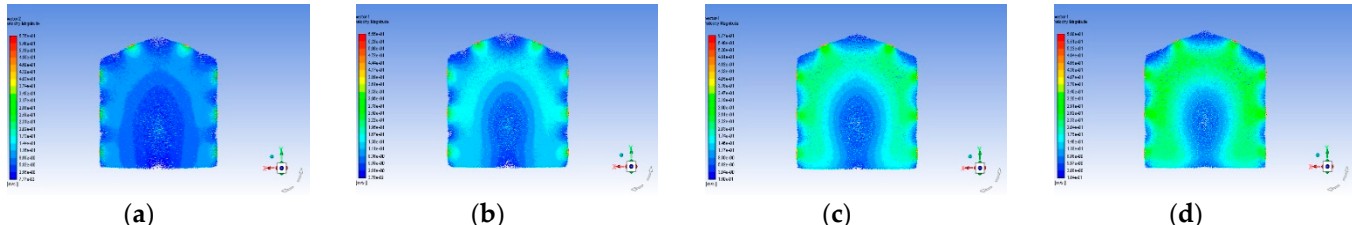

**Figure 6.** Velocity vector model of the guide box with different angles of the guide plate (main view). The angles of the deflector in the picture are (**a**) 150°, (**b**) 160°, (**c**) 170°, and (**d**) 180° from left to right.

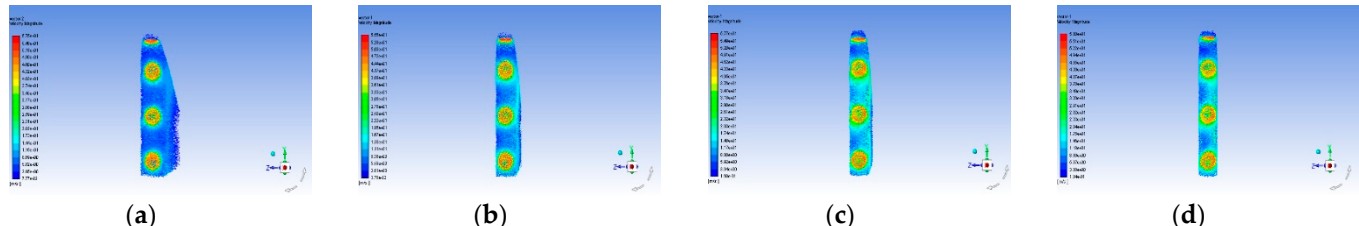

**Figure 7.** Velocity vector model of the guide box with different angles of the guide plate (left view). The angles of the deflector in the picture are (**a**) 150°, (**b**) 160°, (**c**) 170°, and (**d**) 180° from left to right.

**Table 4.** The simulation results of the maximum wind speed of each air outlet under the condition of different angles of the deflector.

| Air Outlet Number | Maximum Wind Speed at Air Outlet (m/s) | | | |
| --- | --- | --- | --- | --- |
| | The Angle of the Deflector Is 150° | The Angle of the Deflector Is 160° | The Angle of the Deflector Is 170° | The Angle of the Deflector Is 180° |
| 1 | 5.75 | 5.55 | 5.77 | 5.80 |
| 2 | 5.46 | 5.28 | 5.49 | 5.51 |
| 3 | 5.18 | 5.00 | 5.20 | 5.22 |
| 4 | 5.75 | 5.55 | 5.77 | 5.80 |
| 5 | 5.75 | 5.55 | 5.77 | 5.80 |
| 6 | 5.18 | 5.00 | 5.20 | 5.22 |
| 7 | 5.46 | 5.28 | 5.49 | 5.51 |
| 8 | 5.75 | 5.55 | 5.77 | 5.80 |
| Range | 0.57 | 0.55 | 0.57 | 0.58 |
| Standard error | 0.237 | 0.228 | 0.236 | 0.240 |

Since the diversion box is symmetrical on both sides, the maximum wind speed value of the air outlet obtained by simulation is also symmetrical. When the angle of the deflector is 160°, the outlet wind speed reaches its minimum, and under other angle conditions, there is a small difference in the wind speed of the air outlet. Compared with the deflector with an included angle of 170°, the maximum wind speed at the air outlet of the deflector with an angle of 150° is similar to it, but from Figure 6, the wind force on the front of the deflector is much smaller than that of the 170° clamp. The angled deflector makes the structure

more stable, and the loss of wind speed in the deflector box is smaller. Considering the structural stability and the air supply effect of the air supply operation department, this study chooses the solution in which the angle of the deflector is 150°.

The air outlets on the deflector box are connected to a side panel with an angle of 75° through a corrugated pipe. The bottom of the panel is connected with a linear push rod motor and a slide rail, so that it has the ability to extend to both sides.

### 2.6. Design of the Automatic Targeting and Position Adjustment System

The main functions of the jet-type orchard remote control spraying machine automatic targeting and orientation adjustment system are to automatically detect the spatial orientation of the fruit tree canopy, adjust the distance from the nozzle to the canopy, and automatically control the opening and closing of the solenoid valve. For the same orchard and the same period, the age and height of the fruit trees were basically the same. Therefore, this study uses remote control to achieve semi-automatic adjustment of the nozzle height and sensor detection data to achieve automatic adjustment of the distance from the nozzle to the canopy.

Since the spraying system on both sides of the spraying machine is an independent unit, the main control chip is 2 STM32 single-chip micro-computers. Each single-chip micro-computer controls four solenoid valves, 4 ultrasonic sensors and 1 linear actuator motor on one side through the IIC bus. Among them, the KS109 ultrasonic sensor is used to automatically target the fruit tree canopy. The detection range is 0.08 m~10 m, the beam angle is approximately 15°, and the detection error is ±5 mm. The distance from the nozzle to the fruit tree canopy is controlled by a linear actuator motor, the rated voltage is 24 V, and the effective stroke is 0~300 mm. To solve the problems of a long delay time in the spraying operation of existing sprayers and a poor target accuracy, this design adopts a JY31N-type acceleration sensor, its acceleration stability is 0.098 N·m, and the vehicle traveling speed is measured by an iterative method. The front sensor method is adopted to make the distance between the nozzle and the nozzle 400 mm. This distance can be accurately delayed in accordance with the traveling speed of the vehicle and reduce the response lag of the sprayer caused by the adjustment of the mechanism's orientation.

Research has been carried out horizontal stratification on the canopy of fruit trees [20], that is, the bottom, middle, top, and canopy areas of the canopy. The ultrasonic sensors on the same side were detected in turn according to the principle of bottom-up. The spatial layout of the ultrasonic sensor is shown in Figure 8.

The literature [27] concluded that the spraying quality is the best when the spray nozzle is 1 m away from the fruit tree canopy. Therefore, in this study, the distance between the spray nozzle and the fruit tree canopy is adjusted based on the detection distance of the bottom sensor. The solenoid valve is controlled on and off. The control scheme is shown in Figure 9. The above-mentioned control scheme is executed after the system is powered on. It ends when all the spraying operations are completed, and the power is cut off. Once the program has an error in the running process, the watchdog program will send a restart signal, and the system will run again. Since the ultrasonic sensors, IIC bus, and other related technologies used in the automatic target and orientation adjustment system are mature, we are no longer testing the system and the sensitivity of each module and other parameters.

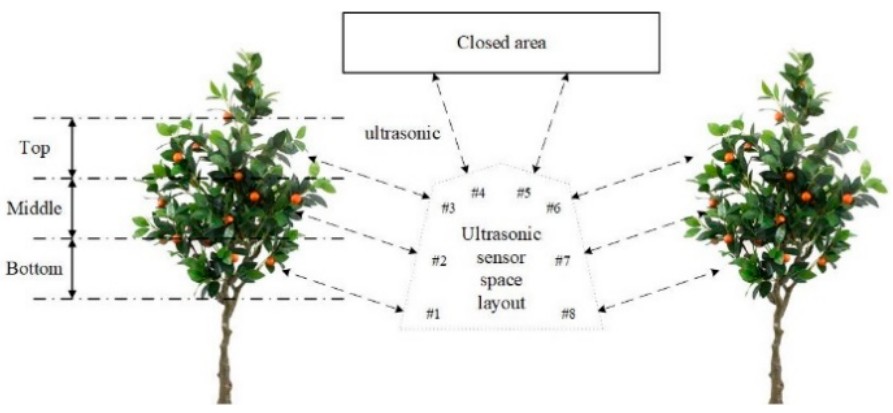

**Figure 8.** Ultrasonic sensor space layout.

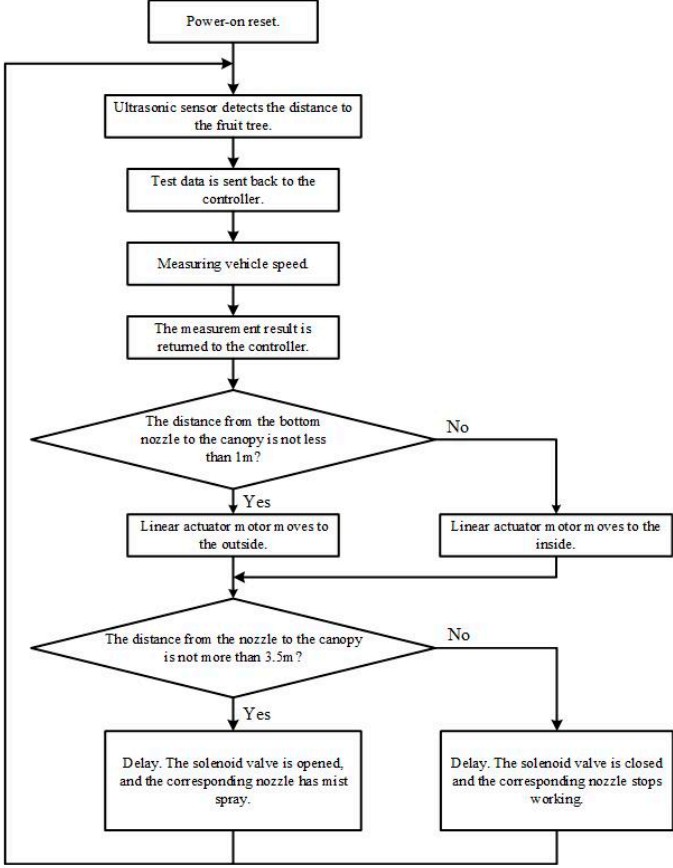

**Figure 9.** System control scheme.

### 3. Results

To test whether each part of the jet orchard remote control sprayer meets the requirements of orchard plant protection, this paper tested the effective range, flow rate, and target effect of the sprayer and three single-factor indexes of the sprayer as well as the curb spray effect. The following experiments were carried out in the laboratory.

### 3.1. Effective Range Test

Since the effective range of the jet-type orchard remote control sprayer is related to the air supply intensity, the outlet wind speed was measured in this study before the test. The measured inlet wind speed is approximately 9.5 m/s, the test equipment is a SMART SENSOR AS856 digital anemometer, and the wind speed values at each outlet are measured, as shown in Table 5.

**Table 5.** Wind speed measurement of air outlet.

| Air Outlet Number | Measured Wind Speed of Air Outlet (m/s) | | | | Standard Deviation | Coefficient of Variation | Simulation Value (m/s) | Measured Error |
|---|---|---|---|---|---|---|---|---|
| | Observation 1 | Observation 2 | Observation 3 | Average (Maximum–Minimum) | | | | |
| 1 | 4.52 | 4.77 | 4.75 | 4.68 (4.77–4.52) | 0.113 | 0.024 | 5.75 | 18.61% |
| 2 | 5.51 | 5.26 | 5.19 | 5.32 (5.51–5.19) | 0.137 | 0.026 | 5.46 | 2.56% |
| 3 | 5.75 | 5.69 | 5.81 | 5.75 (5.81–5.69) | 0.049 | 0.009 | 5.18 | 11.00% |
| 4 | 5.85 | 5.82 | 5.64 | 5.77 (5.85–5.64) | 0.093 | 0.016 | 5.75 | 0.35% |
| 5 | 5.83 | 5.65 | 5.77 | 5.75 (5.83–5.65) | 0.075 | 0.013 | 5.75 | 0.00% |
| 6 | 5.48 | 5.67 | 5.89 | 5.68 (5.89–5.48) | 0.168 | 0.029 | 5.18 | 9.65% |
| 7 | 5.57 | 5.41 | 5.40 | 5.46 (5.57–5.40) | 0.078 | 0.014 | 5.46 | 0.00% |
| 8 | 4.99 | 4.61 | 4.59 | 4.73 (4.99–4.59) | 0.184 | 0.039 | 5.75 | 17.74% |

The measured wind speed values in the Table 4 are basically close to the simulation values, but there are differences between the measured results and the simulation ones in terms of wind speed values of the No. 1 and No. 8 air outlets, which has a certain relationship with the solenoid valve installed inside the diversion box. Two special cases are observed from the actual measurement error column. The measured value of the wind speed of No. 1 and No. 8 outlets is less than the simulated value, and the measured value of the wind speed of No. 3 and No. 6 outlets is greater than the simulated value. After analysis, we discovered this situation is caused by errors in the processing of the deflector box, and the center of the front deflector is not directly facing the center of the fan. In addition, the solenoid valve and pipeline at the inside of the diversion box also have a certain influence on the internal flow field.

The effective range of the jet orchard remote control sprayer is determined according to the effective range measurement method given in the *JB/T 9782-2014 General Test Method for Plant Protection Machinery*. This test is carried out in an environment without natural wind. Each solenoid valve is connected to a 24 V DC power supply in turn, and the sprayer is started. Measurement begins when the air supply unit is operating stably and the pressure of the triplex plunger pump is stable at 1.6 MPa, as shown in Table 6.

**Table 6.** Effective range measurement results.

| Nozzle Number | Effective Range Measurement (m) | Nozzle Number | Effective Range Measurement (m) |
|---|---|---|---|
| 1 | 3.52 | 5 | 3.61 |
| 2 | 3.63 | 6 | 3.76 |
| 3 | 3.74 | 7 | 3.69 |
| 4 | 3.66 | 8 | 3.58 |
| Effective range mean (m) | | 3.65 | |

In summary, the effective range of the jet remote control sprayer for orchards is 3.65 m, which is greater than the design value of 3.5 m. It can handle the spraying operations of fruit trees with a larger row spacing, so it meets the design requirements.

### 3.2. Maximum Flow Rate Determination Test

The volume of the medicine box used in this research was 300 L, and there was a scale for every 50 L. Before the test, all solenoid valves were connected to a 24 V DC power supply, 300 L of water was poured into a medicine box, and the sprayer was started; after waiting until the water volume dropped to 250 L, the time used was calculated, and the ratio of the water volume change to the time was considered the maximum value flow rate. Then, the maximum flow rate of the sprayer was measured, as shown in Table 7.

**Table 7.** Flow velocity measurement result.

| Number of Test | Change in Water Quantity (L) | Test of Time (s) | Maximum Velocity (L/min) |
|---|---|---|---|
| 1 | 50 | 486 | 6.17 |
| 2 | 50 | 473 | 6.34 |
| 3 | 50 | 492 | 6.10 |

In summary, the maximum flow rate range of the sprayer is between 6 and 6.5 L/min, and the test data fluctuated. In addition to the small error in the remaining water reading, it is related to the stability of the three-cylinder plunger pump.

*3.3. Target Effect Determination Test*

To test the effect of the jet-type orchard remote control sprayer on the target, the travel speed of the sprayer was set to 1 m/s, and the detection target was a rectangular cardboard with a length of 2 m and a width of 0.5 m. From the detection of the target by the ultrasonic sensor to the automatic opening of the solenoid valve, the distance difference between the nozzle and the targeting in the direction of travel was measured, as shown in Table 8.

**Table 8.** The measured result of the targeting effect. When the distance difference between the sprinkler head and the targeting object in the direction of travel is positive, it means that the solenoid valve is opened in advance, and the sprinkler head does not go to the specified position. When the distance difference is negative, the solenoid valve is delayed opening, and the nozzle has exceeded the specified position.

| Number of Test | Measured Distance between Sprinkle Head and Targeting Object (m) | The Distance Difference between the Nozzle and the Object in the Direction of Travel (mm) |
|---|---|---|
| 1 | 0.5 | 52 |
| 2 | 0.5 | 44 |
| 3 | 0.5 | 53 |
| Average | | 50 |
| 4 | 2 | 38 |
| 5 | 2 | 31 |
| 6 | 2 | 35 |
| Average | | 35 |
| 7 | 3.5 | −9 |
| 8 | 3.5 | −12 |
| 9 | 3.5 | −7 |
| Average | | −9 |

In summary, the farther the jet-type orchard remote control sprayer is from the targeting, the smaller the distance difference between the nozzle and the targeting in the direction of travel. The root cause of the above problems is the error in the accuracy of vehicle speed detection. To enable the ultrasonic sensor to perform chemical spraying operations in a timely and accurate manner after automatically detecting the target, this design increases the distance between the sensor and the spray head through the sensor front method so that the control system can adapt to the current vehicle speed, and the delay is satisfied. However, the real-time measurement method of vehicle speed used in this design has a certain error, which leads to the target error of the sprayer with a maximum value of 50 mm. However, the error is relatively small compared with the canopy size and spray width of fruit trees and can be ignored.

### 3.4. Spray Effect Test

3.4.1. Test Plan Design

The test was conducted in the Modern Agricultural Equipment Laboratory of the College of Engineering and Technology of Southwest University. The test time was 14 April 2021, and the environmental wind speed was 0.35 m/s. The test bench was constructed by employing aluminum profiles, branches were hung on each test level, and the water-sensitive test paper was placed at appropriate points. The size of the test bench and the water-sensitive test paper points are shown in Figure 10.

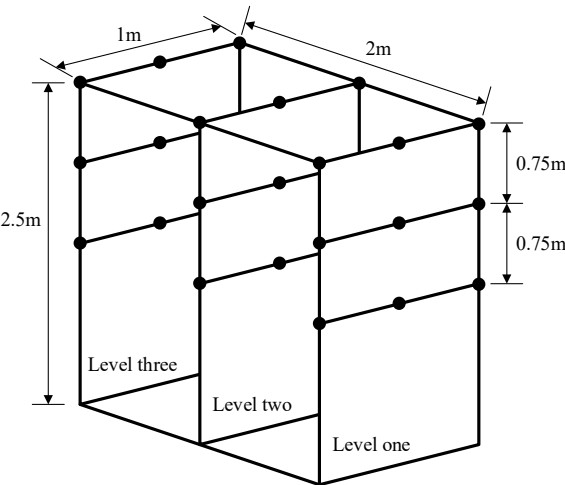

**Figure 10.** Test bench size and water-sensitive test paper point map. The black dots in the figure are the water-sensitive test paper points, and the water-sensitive test paper is hung on the front and back of each point.

After completing the test bench layout according to the above points, the jet orchard sprayer was set, and the spray height was adjusted so that all water-sensitive test paper test points fell within the spray range. We scanned the water-sensitive test paper after the test and imported it into the software to obtain related data. Since the test strip analysis software is a mature technology, the specific operation method will not be repeated in this article. The test site is shown in Figure 11.

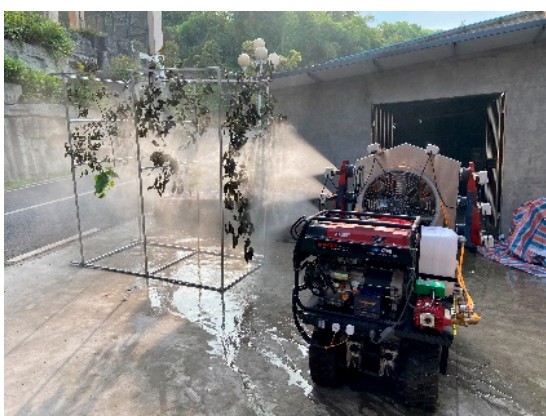

**Figure 11.** Test site.

3.4.2. Analysis of Test Results

The jet orchard sprayer maintained a vehicle speed of 1.5 m/s, the fan rotated at 1400 r/min to drive through the test bench, the water-sensitive test paper was removed and scanned, and the image was processed and analyzed in the test paper analysis software.

The above operation was repeated three times, and the relevant parameters of the spray effect of the jet orchard sprayer were obtained, as shown in Table 9.

**Table 9.** Related parameters of spray effect test.

| Number of Test | Tested Positions | 10% Diameter by Volume (μm) | Volume Median Diameter (μm) | 90% Diameter by Volume (μm) | Total Number of Droplet (Drops) | Sampling Area (cm²) | Deposition Density (Drops/cm²) | Coverage Rate |
|---|---|---|---|---|---|---|---|---|
| 1 | Front of level 1 | 104.14 | 220.58 | 783.26 | 397 | 6.17 | 64.34 | 25.16% |
|  | Back of level 1 | 87.20 | 132.30 | 785.48 | 57 | 5.15 | 11.07 | 1.28% |
|  | Front of level 2 | 93.26 | 155.68 | 360.88 | 719 | 6.76 | 106.36 | 16.46% |
|  | Back of level 2 | 87.20 | 128.48 | 275.16 | 13 | 5.85 | 2.22 | 0.16% |
|  | Front of level 3 | 101.54 | 141.60 | 198.08 | 134 | 4.52 | 29.65 | 7.27% |
|  | Back of level 3 | 83.96 | 93.26 | 134.16 | 5 | 6.38 | 0.78 | 0.03% |
| 2 | Front of level 1 | 112.57 | 231.13 | 786.17 | 409 | 6.29 | 65.02 | 26.17% |
|  | Back of level 1 | 86.45 | 145.62 | 783.89 | 63 | 5.51 | 11.43 | 1.65% |
|  | Front of level 2 | 89.33 | 150.10 | 358.70 | 721 | 6.83 | 105.56 | 15.33% |
|  | Back of level 2 | 87.31 | 114.08 | 281.32 | 15 | 5.97 | 2.51 | 0.25% |
|  | Front of level 3 | 103.50 | 144.95 | 201.29 | 142 | 5.13 | 27.68 | 9.29% |
|  | Back of level 3 | 79.66 | 88.51 | 134.12 | 7 | 5.99 | 1.17 | 0.09% |
| 3 | Front of level 1 | 107.60 | 227.84 | 787.33 | 386 | 5.94 | 64.98 | 25.45% |
|  | Back of level 1 | 90.47 | 135.25 | 780.65 | 52 | 4.87 | 10.68 | 1.17% |
|  | Front of level 2 | 94.15 | 160.60 | 353.72 | 716 | 6.52 | 109.82 | 18.74% |
|  | Back of level 2 | 85.28 | 121.33 | 277.31 | 10 | 5.50 | 1.82 | 0.22% |
|  | Front of level 3 | 99.72 | 128.96 | 195.60 | 126 | 4.18 | 30.14 | 8.61% |
|  | Back of level 3 | 83.11 | 90.76 | 130.08 | 8 | 5.85 | 1.37 | 0.13% |

The test data showed that, under the action of wind, the pesticide droplets sprayed by the jet orchard remote control sprayer had a certain penetration ability, and the smaller the droplet diameter was, the stronger the penetration ability. Considering the front spraying effect, most of the large-diameter droplets were attached to the first layer. More than half of the droplets were attached to the second layer, but due to the overall small droplet diameter, the coverage rate was lower than that of the first layer. The parameters of the third layer were significantly reduced but still met the requirements of low-volume spraying in the orchard. From the perspective of the spraying effect on the back side, droplets were attached to each layer, but as the depth of the canopy increased, the number of droplets, density, and other indicators attenuated more severely. The pesticide droplets in each layer of fog drops attached to the leaves were not condensed into water droplets to fall, which effectively solved the problem of pesticides infiltrating the soil and causing soil compaction.

To further analyze the performance indicators of the sprayer, this study selects the droplet deposition density parameter in the test table as the main indicator of sprayer performance evaluation. Compared with the droplet deposition rate, it can clearly reflect the value and distribution relationship, according to the test results. According to the above method, we calculated the average value of the deposition density of each layer on the front and back of each test and the total average value of the deposition density of each layer, calculated the standard error, and used the t-test to obtain the P-value. The data analysis of the droplet deposition density is shown in Table 10.

As can be seen from the table, there is no significant difference in the data of each group. In order to further verify the spraying effect of the jet-type orchard remote control sprayer, this study compares it with the sprayer described in [20] from the indicator of the droplet deposition density that can directly reflect the number of droplets per unit area. Because both belong to the jet spraying machine, there are only certain differences in the structure of the air supply system and the structure of the spraying device. Therefore, the two are comparable. This paper compares the average deposition density of the two models before and after each layer of fruit trees and gives a distribution map of the percentage of each type of deposition density of each layer in the total deposition density, as shown in Figures 12 and 13.

**Table 10.** Data analysis.

| Tested Positions | Number of Test | Actual Deposition Density (Drops/cm$^2$) | Average Deposition Density (Drops/cm$^2$) | Standard Deviation | *p*-Value |
|---|---|---|---|---|---|
| Level 1 | 1<br>2<br>3 | 37.71<br>38.23<br>37.83 | 37.92 | 0.223 | 0.9921 |
| Level 2 | 1<br>2<br>3 | 54.29<br>54.04<br>55.82 | 54.72 | 0.786 | 0.9940 |
| Level 3 | 1<br>2<br>3 | 15.21<br>14.42<br>15.76 | 15.13 | 0.547 | 0.9969 |

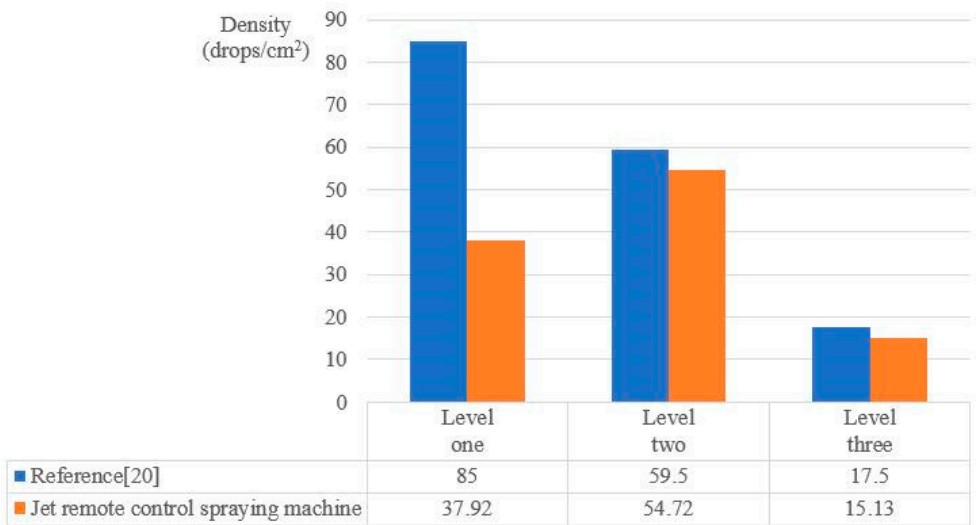

**Figure 12.** Comparison chart of droplet deposition density.

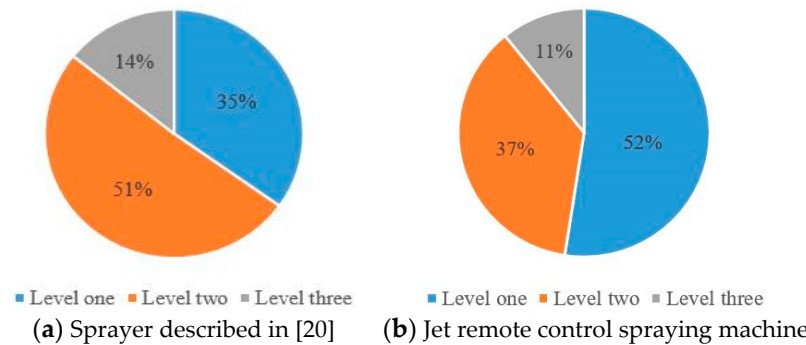

(**a**) Sprayer described in [20]     (**b**) Jet remote control spraying machine

**Figure 13.** The percentage of droplet deposition density in each layer of the fruit tree canopy.

The results in the figure show that, although the deposition density of the sprayer mentioned in [20] is better than that of the sprayer designed in this study, more than half of the droplets are concentrated on the surface of the fruit tree canopy. It is well known that fruit tree diseases and insect pests often gather in the wet places inside the canopy of fruit trees, which requires that the density of droplets in the canopy of fruit trees should be greater than the density of droplets on the surface of the canopy. Since the air supply system provided in this study adopts the structure of a diversion box, the wind speed at each outlet is evenly distributed, and it can assist the droplets to penetrate the fruit tree canopy. Therefore, the second layer of fruit tree canopy has a high droplet deposition density. From the test results, in the spray effect test of the jet orchard sprayer, 63% of the droplets were absorbed in the fruit tree canopy, which basically solved the problem of weak droplet penetration.

## 4. Discussion

At present, mechanized spraying suffers problems such as the small amount of drug deposition inside the canopy and the low accuracy of automatic targeting of the spraying equipment. The main reasons for the small amount of drug deposition inside the canopy are the failure to form an effective air supply operation, the large air outlet of the sprayer, the poor air uniformity and the obvious attenuation of the external flow field under actual working conditions [3,4,7]. The machine concentrates all the air volume in the sealed diversion box, redistributes the air volume of each outlet under stable wind pressure conditions, increases the air volume of each outlet, and ensures the stability and directionality of the external flow field. Thus, it can effectively assist the droplets in penetrating the canopy. The problem of low accuracy spraying equipment on automatic targeting is mainly because the complex sensor ranging system improves the accuracy of measurement data while ignoring the timeliness of data processing, which in turn leads to delays in the opening and closing of the solenoid valve [19]. In previous research (e.g., [20]), the method of the multi-sensor cooperative operation was adopted to realize the hybrid ranging of ultrasonic sensors and laser diode arrays and to correlate the distance detection data with the real-time speed data of the vehicle. However, the lagging response of the solenoid valve reduced the accuracy of the target. Therefore, this research simplifies the distance measurement sensor system and only uses ultrasonic sensors to determine the distance and guide the orientation adjustment. The test data show that this scheme can improve the response time of the automatic targeting system and the accuracy of the target.

This research has achieved the purpose of improving the spraying quality and spraying efficiency of standardized orchards in hilly and mountainous areas, reducing the labor intensity of plant protection workers, and alleviating soil compaction and environmental pollution caused by excessive spraying. Although the jet-type orchard remote control sprayer fails to achieve the technical goals of online mixing [29] and variable spraying [30], the limit spraying volume of the spraying machine in 1 h does not exceed 390 L and thus can be considered low-volume spraying. In addition, sprayed pesticide droplets have a certain penetration ability, which can basically meet the daily spraying work of standardized orchards in hills and mountains. The jet-type orchard remote control sprayer has an effective range of no more than 3.5 m and a maximum flow rate range of 6~6.5 L/min in a static state, achieving the goals of low-volume spraying. Within the effective range, the farther the nozzle is from the fruit tree, the higher the accuracy of the automatic targeting. When the vehicle speed is 1.5 m/s and the fan speed is 1400 r/min, the sprayer can complete the low-volume spraying operation of fruit trees within 4 m of a canopy diameter. The coverage rate of pesticide droplets on the surface of the canopy profile is 25.16%, the amount of deposition is 1.9437, the coverage rate of the back is 1.28%, and the amount of deposition is 0.0634. When the canopy depth is 1 m, the front coverage rate of pesticide droplets is 16.46%, the deposition amount is 0.8147, the back coverage rate is 0.16%, and the deposition amount is 0.0041. When the canopy depth is 2 m, the front coverage rate of pesticide droplets is 7.27%, the deposition amount is 0.3236, the back coverage rate is 0.03%, and the deposition amount is 0.0005. As the depth of the canopy increases, both the positive and negative coverage rates of the droplets are attenuated. Especially when the depth of the canopy is 2 m, the number of pesticide droplets on the back is extremely small, and it is not ruled out that the pesticide droplets on the back at this depth are caused by drifting fog. Comprehensive analysis of various test data shows that the coupling relationship between wind pressure and air volume has a certain impact on the coverage and deposition of droplets. Increasing the air volume by technical means may cause problems such as increased droplet drift and mechanical damage to branches and leaves.

## 5. Conclusions

This research device fulfills the basic requirements for automatic targeting and spraying of standardized orchards in hilly and mountainous areas and improves the uniformity

and directionality of the external flow field of the air supply system. However, the influence of the operation stability of the fan and three-cylinder plunger pump on the spraying effects of the sprayer has not been resolved, and the prototype has not been further analyzed and optimized through field trials. In the future, the power source will be changed to improve stability, image processing technology will be used to detect and analyze the types and degrees of fruit tree diseases and insect pests, field trials will be conducted to obtain data, and the sprayer structure will be optimized to further improve the spraying quality of fruit trees.

**Author Contributions:** Conceptualization, C.M. and G.L.; methodology, G.L.; software, C.M.; validation, C.M., G.L. and Q.P.; formal analysis, C.M.; investigation, Q.P.; resources, C.M.; data curation, C.M. and Q.P.; writing—original draft preparation, C.M.; writing—review and editing, G.L.; visualization, C.M.; supervision, Q.P.; project administration, G.L.; funding acquisition, C.M. and Q.P. All authors have read and agreed to the published version of the manuscript.

**Funding:** This research was funded by Southwest University's special fund project for basic scientific research business expenses under grant #XDJK2020C030 and major scientific and technological innovation projects in Shandong Province under grant #2019JZZY0a20623.

**Institutional Review Board Statement:** Not applicable.

**Informed Consent Statement:** Not applicable.

**Data Availability Statement:** Not applicable.

**Conflicts of Interest:** The authors declare no conflict of interest.

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
