# Peer review of "Design and Test of a Jet Remote Control Spraying Machine for Orchards"

_agriengineering, doi:10.3390/agriengineering3040050_

Round 1

Reviewer 1 Report

The authors design a device for solving a not enough automation and/or efficiency of the current automated machines for the pesticides application. However:

  1. It is not specified for what kind of orchard is designed the device (for example, the size of the trees, distance between trees or the planted species). I suggest indicating the orchards characteristics were the device could be used.
  2. It should be also clear in the methods that all tests were carried out in laboratory and that this is a first step of the design.
  3. The number of observations (only three) performed for the analyzed parameters is too low. More observations should be included for accurate conclusions (this is the major flow of the work).

Some specific comments:

The abstract generally summarize the work, however, it needs a revision of the English, for example:

Line 9. “Aimed at issues associated with the poor air supply and poor automatic targeting accuracy of existing orchard sprayers…” Please, revise the English

Line 11-14. I suggest modifying the structure of the sentence: “The uniformity of deposition is improved by optimizing…”

Revise also this part of the sentence: “a uniform wild field distribution is realized”

Line 14-15.Similarly, I suggest “an accurate positioning of the fruit tree canopy space orientation is achieved through automatic targeting and azimuthal adjustment systems”

Line 15. “the solenoid valve is controlled to be turned on and off” I suggest to explain here in which situation is turned on or off.

Line 34. Please, specify what do you mean by “ bumps”, why the driver  can be easily fatigued? Do you mean that the process is not enough automated?

Line 63-66. Revise the English. Also, add here that the parameters were tested in laboratory.

Line 99-100. Do you mean in south China? Slopes in other parts of the world can be frequently higher. Please, specify.

246-247. “the requirements of orchard plant protection” This is very specific of each kind of orchard. Additionally, tests were done in laboratory (and not yet in field) and this should be clear across the manuscript and in the abstract.

Line 258. “but there is a certain difference between the air outlets at the bottom of both sides and the simulation value” which are the outlets at the bottom in the table? Should not be this “certain difference” confirmed with a statistical analysis? Additionally, three observations is too low, it is not possible adding more observations and then present the average and standard error (idem for the other tested parameters across the manuscript.

Table 4. I suggest showing the average (maximum -minimum), or better, average (standard error) in the table, and specifying in the text that the measure was performed three times (better if more observations can be included). Idem for the other results.

Author Response

Response to Reviewer 1 Comments

Dear reviewer:

        Thank you for your valuable suggestions for my article. Our research team conducted a deep analysis and thinking on your suggestions, and revised the relevant content. The modified content has been marked in the article in red font. In addition, we have explained some issues in this document. Please review it.

Point 1:

        It is not specified for what kind of orchard is designed the device (for example, the size of the trees, distance between trees or the planted species). I suggest indicating the orchards characteristics were the device could be used.

Response 1:

        The spraying machine mentioned in this research is a universal machine, especially suitable for citrus orchards, vineyards and other orchards that grow dwarf varieties. In response to your question, after discussion by the research group, a related expression, related content has been added to lines 80~86.

Point 2:

        It should be also clear in the methods that all tests were carried out in laboratory and that this is a first step of the design.

Response 2:

        It has been supplemented in lines 76~78 and marked in red font.

Point 3:

        The number of observations (only three) performed for the analyzed parameters is too low. More observations should be included for accurate conclusions (this is the major flow of the work).

Response 3:

        Based on your comments, the research team conducted a discussion. We believe that usually 3 repeated experiments are the basic operation, and the experiments involved in this article are all confirmatory, and the interaction between various variables is not discussed. Therefore, We finally confirmed 3 trials. Of course, we will adopt your suggestions in subsequent reports on the relevant content of field trials, increase the number of experiments, and investigate the correlation between the variables.

Point 4:

        Line 9. “Aimed at issues associated with the poor air supply and poor automatic targeting accuracy of existing orchard sprayers…” Please, revise the English.

        Line 11-14. I suggest modifying the structure of the sentence: “The uniformity of deposition is improved by optimizing…”Revise also this part of the sentence: “a uniform wild field distribution is realized”

        Line 14-15.Similarly, I suggest “an accurate positioning of the fruit tree canopy space orientation is achieved through automatic targeting and azimuthal adjustment systems”

Response 4:

        The modification has been completed in lines 9~15 and marked in red font.

Point 5:

        Line 15. “the solenoid valve is controlled to be turned on and off” I suggest to explain here in which situation is turned on or off.

Response 5:

        Corresponding explanations have been added in lines 15~16 and marked in red font.

Point 6:

        Line 34. Please, specify what do you mean by “ bumps”, why the driver  can be easily fatigued? Do you mean that the process is not enough automated?

Response 6:

        Corresponding explanations have been added in lines 39~40 and marked in red font.

Point 7:

        Line 63-66. Revise the English. Also, add here that the parameters were tested in laboratory.

Response 7:

        The modification has been completed in lines 69~75 and marked in red font.

Point 8:

        Line 99-100. Do you mean in south China? Slopes in other parts of the world can be frequently higher. Please, specify.

Response 8:

        According to surveys, the slopes of hills and mountains in China are generally 10°~25°. Of course, there are some higher slopes, but this kind of slope generally does not use crawler-type equipment for plant protection work. Therefore, the modification has been completed in line 116 and marked in red font.

Point 9:

        Line 246-247. “the requirements of orchard plant protection” This is very specific of each kind of orchard. Additionally, tests were done in laboratory (and not yet in field) and this should be clear across the manuscript and in the abstract.

Response 9:

        Corresponding explanations have been added in line 277 and marked in red font.

Point 10:

        Line 258. “but there is a certain difference between the air outlets at the bottom of both sides and the simulation value” which are the outlets at the bottom in the table? Should not be this “certain difference” confirmed with a statistical analysis? Additionally, three observations is too low, it is not possible adding more observations and then present the average and standard error (idem for the other tested parameters across the manuscript.

        Table 4. I suggest showing the average (maximum -minimum), or better, average (standard error) in the table, and specifying in the text that the measure was performed three times (better if more observations can be included). Idem for the other results.

Response 10:

        According to the questions you raised, the research team considered that Table.4 reflects a kind of confirmatory experimental results, and there is no need to explore the change law of the wind speed of each outlet. Therefore, we did not increase the measurement data. As for the increase in the max-min, average error and standard error you mentioned, we have supplemented the corresponding content. After analyzing and thinking about the result, it has been added in lines 285~294 and marked in red font.

Kind regards,

Chi Ma

College of Mechanical and Technology, Southwest University, China

Guanglin Li

College of Mechanical and Technology, Southwest University, China

Qiangji Peng

Academy of Agricultural Machinery Science in Shandong, China

Reviewer 2 Report

MS is interesting; it is however missing critical elements that are required in a scientific MS:

Materials and Methods section must include:

  • enough information to allow reader to duplicate study
  • a clear description of the statistical experimental design, # of replicates performed, any blocking variables, what was used as a control to assess performance, etc?  Describe which statistical tests will be conducted to assess experimental results.
  • on simulation:
    • dimensional drawings are required so reader can duplicate/extend this work
    • boundary conditions for simulation must be stated
    • type of solver used in simulation must be specified
    • type of flow modeled (laminar or turbulent; Reynolds #), was a turbulence model utilized (if so which one?)

Discussion/Results: must include results of statistical analysis:

  • p-values
  • class groupings between treatments (are they statistically different (i.e. significant)?

Author Response

Response to Reviewer 2 Comments

Dear reviewer:

Thank you for your valuable suggestions for my article. Our research team conducted a deep analysis and thinking on your suggestions, and revised the relevant content. The modified content has been marked in the article in green font. In addition, we have explained some issues in this document. Please review it.

Point 1:

Materials and Methods section must include:

Enough information to allow reader to duplicate study a clear description of the statistical experimental design, # of replicates performed, any blocking variables, what was used as a control to assess performance, etc?  Describe which statistical tests will be conducted to assess experimental results.

Response 1:

Corresponding expressions have been added to lines 264~270 and marked in green font.

Point 2:

Dimensional drawings are required so reader can duplicate/extend this work.

Response 2:

Based on your suggestions, we added the diversion box size chart in lines 205~207.

Point 3:

Boundary conditions for simulation must be stated.

Type of solver used in simulation must be specified.

Type of flow modeled (laminar or turbulent; Reynolds #), was a turbulence model utilized (if so which one?).

Response 3:

According to this question, the boundary conditions, type of solver used in simulation, and type of flow modeled have been added in lines 208~210 of the article, and marked in green font.

Point 4:

Discussion/Results: must include results of statistical analysis:

p-values

class groupings between treatments (are they statistically different (i.e. significant)?

Response 4:

This research focuses on the design of the spraying machine. Through laboratory experiments, it is verified that the equipment can achieve a certain spraying effect. The investigation of the interaction between wind speed, vehicle travel speed and spraying effect needs to be tested in the field. In addition, take the spraying effect experiment as an example. Although the experiment is carried out in a laboratory, there are many levels and parameters measured, especially the parameters are in a parallel relationship. Under the conditions of different levels and different indicators, the significance evaluation cannot be carried out. Therefore, the significance test was not performed. However, this is part of our follow-up work and will be reported separately.

Kind regards,

Chi Ma

College of Mechanical and Technology, Southwest University, China

Guanglin Li

College of Mechanical and Technology, Southwest University, China

Qiangji Peng

Academy of Agricultural Machinery Science in Shandong, China

Reviewer 3 Report

The paper contains an interesting topic that is extremely important for agriculture. A great value of this paper is the practical analyses of spraying machine applications carried out by the authors. Overall the paper is well written albeit I have a few minor comments:

Work to refine the purpose of the paper and include it in the introduction paragraph.

Please consider expanding the introductory paragraph to emphasize why this topic is important to agricultural practice and science.

Figure 2 is too small and is unreadable. Please increase the size of the figure and the font size of the descriptions.

The same for figures 7 and 8 - please increase the font.

In Table 8, superscript at 2 should be used when writing the unit cm2.

Author Response

Response to Reviewer 3 Comments

Dear reviewer:

Thank you for your valuable suggestions for my article. Our research team conducted a deep analysis and thinking on your suggestions, and revised the relevant content. The modified content has been marked in the article in blue font. In addition, we have explained some issues in this document. Please review it.

Point 1:

Work to refine the purpose of the paper and include it in the introduction paragraph.

Response 1:

It has been supplemented in lines 31~35 and marked in blue font.

Point 2:

Please consider expanding the introductory paragraph to emphasize why this topic is important to agricultural practice and science.

Response 2:

It has been supplemented in lines 68~74 and marked in blue font.

Point 3:

Figure 2 is too small and is unreadable. Please increase the size of the figure and the font size of the descriptions.

The same for figures 7 and 8 - please increase the font.

Response 3:

This question has been revised in the article.

Point 4:

In Table 8, superscript at 2 should be used when writing the unit cm2.

Response 4:

The question has been modified in the original form and marked in blue font.

Kind regards,

Chi Ma

College of Mechanical and Technology, Southwest University, China

Guanglin Li

College of Mechanical and Technology, Southwest University, China

Qiangji Peng

Academy of Agricultural Machinery Science in Shandong, China

Round 2

Reviewer 1 Report

Generally the suggestions and comments were followed or clarify.

A couple of comments:

  1. I believe my comment about the table 4 (and results) was misunderstood. My previous comment was:

“Table 4. I suggest showing the average (maximum -minimum), or better, average (standard error) in the table, and specifying in the text that the measure was performed three times (better if more observations can be included). Idem for the other results.”

I meant that is probably awkward  to show data as “the first time, the second time, the third time” and it would be more elegant to show the average of the observed values and within brackets the maximum and minimum values, for example in the air outlet number 1: 4.68 (4.77-4.52). Alternatively (not simultaneously), the average and within brackets the standard error -i.e., 4.68 ± “the value of the standard error”- would be more explanatory, but this second option would be meaningful if more than three observations were performed, what is not the case. Also, the heading “Analysis” for those numbers does not seem appropriated, because any analysis (such an ANOVA to confirm if the differences are significant or not) was performed. Finally, in the current form, it is not clear what the max-min column is referring to. Therefore, and given than only three observations were done, the original table may be maintained but with the headings: “observation 1; observation 2; observation 3).

  1. A revision by a native English would improve the manuscript.

Reviewer 2 Report

MS is improved over last version but is still missing key points;

  • MS needs a lot more detail on the CFD model was setup; size of mesh; type of mesh (structured or ?), what boundary conditions were used, etc.  Reader needs to be able to duplicate this work; as it sits, this is not possible from the scant description of how the model was setup. 
  • Information needs to be provided as to why the particular Reynold's # was selected, it seems very low bordering on laminar yet for this type of application is likely much higher... so more detail needs to support this section.
  • MS is missing experimental statistical design explanation; it appears from presented results that 3 replicates were done; and a very rudimentary analysis comparing error between simulation and real system which is not sufficient nor of much interest.  MS should zero in on the relevant results and run the statistics ANOVA between EXPERIMENTAL tests (which aren't clear in the MS).  MS jumps from section 2.5 simulation to 2.6 
  • authors discuss optimal baffle angle as judge from the CFD model; however they need to provide more description on how they make that determination from the CFD model.

Round 3

Reviewer 2 Report

Authors have failed to address the concerns of the previous reviews and MS is still lacking in basic statistical experimental designs and statistical analysis.  There is a lack of control comparison for the experiments and the experiments that are described are not analyzed following standard statistical methods.  As such the MS, in its current form, is better suited as a submission as a technical note; however to do so authors will need to add enough information such that a reader could build and duplicate the apparatus that MS reports on.

Authors are encouraged to address deficiencies discussed above and resubmit MS as either a Technical Note, with full documentation on how to build and construct device, or as an Article with additional documentation on statistical methods and analysis with proper comparison to control that will provide some substance as to the level of performance improvement that this design seeks to address.   
